# Simulation of Total Ionizing Dose Effects Technique for CMOS Inverter Circuit

**DOI:** 10.3390/mi14071438

**Published:** 2023-07-18

**Authors:** Tianzhi Gao, Chenyu Yin, Yaolin Chen, Ruibo Chen, Cong Yan, Hongxia Liu

**Affiliations:** Key Laboratory for Wide Band Gap Semiconductor Materials and Devices of Education, School of Microelectronics, Xidian University, Xi’an 710071, China

**Keywords:** total ionizing dose effect, FDSOI devices, inverter

## Abstract

The total ionizing dose (TID) effect significantly impacts the electrical parameters of fully depleted silicon on insulator (FDSOI) devices and even invalidates the on–off function of devices. At present, most of the irradiation research on the circuit level is focused on the single event effect, and there is very little research on the total ionizing dose effect. Therefore, this study mainly analyzes the influence of TID effects on a CMOS inverter circuit based on 22 nm FDSOI transistors. First, we constructed and calibrated an N-type FDSOI metal-oxide semiconductor (NMOS) structure and P-type FDSOI metal-oxide semiconductor (PMOS) structure. The transfer characteristics and trapped charge distribution of these devices were studied under different irradiation doses. Next, we studied the TID effect on an inverter circuit composed of these two MOS transistors. The simulation results show that when the radiation dose was 400 krad (Si), the logic threshold drift of the inverter was approximately 0.052 V. These results help further investigate the impact on integrated circuits in an irradiation environment.

## 1. Introduction

Power electronics technology is widely used in the aerospace industry and other fields. Owing to the influence of irradiation, the electrical performance of electronic devices, which operate in an irradiation environment, are disturbed and degraded, resulting in the failure of the entire integrated circuit system. The cumulative effect of the electrical performance degradation of electronic devices is caused by high-energy rays and charged particles, that is, the total ionizing dose (TID) effect [1,2,3], which is the main factor affecting the life of electronic devices and cannot be eliminated. Owing to the presence of a buried oxygen (BOX) layer, FDSOI technology separates the substrate layer from the top silicon layer, resulting in good radiation resistance [4,5,6]. Therefore, FDSOI devices are used as the basic research objects. At present, most circuit-level radiation studies focus on the single particle effect, and few studies have investigated the TID effect. Inverters are widely used as the most basic MOS circuit structures [7,8]. Therefore, analyzing the impact of the total dose effect, in integrated circuit systems, which is based on the TID effect of inverter circuits is important.

## 2. Device Structure and Calibration

### Device Structure

The 3D simulation models of the 22 nm fully depleted silicon on insulator (FDSOI) NMOSFET and PMOSFET were established using Sentaurus software, as shown in Figure 1. The source, drain, N well, and P well of the devices were doped with Gaussian doping, and the device substrate was uniformly doped. Some process parameters, such as the doping concentration in the active region and well depth, are the core secrets of the foundry. Therefore, during the building of the device model, setting some of the process parameters through empirical values and then comparing and calibrating them with the electrical characteristic curve measured in the experiment was necessary to finally determine the device parameters.

These devices w all based on 22 nm process nodes. The gate length was 22 nm, and the equivalent oxide thickness (EOT) was 20 Å. The gate width of devices (a) and (b) were 160 nm and 220 nm. A 10 nm top silicon layer was just above the BOX layer, and the spacer width was 10 nm. The BOX layers of devices (a) and (b) were 20 nm SiO_2_.

To ensure the accuracy of the established model, a B1500 semiconductor tester was used to test the saturation zone and linear zone transfer characteristic curves of the nonirradiated 22 nm FDSOI. The test device diagram is shown in Figure 2. The specific testing conditions were as follows: The drain voltage of the NMOS device was set to 0.8 V and 0.1 V, and the gate voltage scanning range was set to 0~1.0 V to test changes in the drain current. Similarly, the drain voltage of the PMOS device was set to −0.8 V and −0.1 V, and the gate voltage scanning range was set to −1.0~0 V to test changes in the drain current.

As the doping concentration and specific process of the device are confidential documents of the factory, based on the layout file, as shown in Figure 3, we can understand the size of the device and use this as a basis to obtain the structural model of the device in Technology Computer Aided Design (TCAD) software through structural modeling, as shown in Figure 1. To further obtain a more accurate physical model, we needed to modify and calibrate the device’s work function, doping the concentration and mobility in the source and drain regions. Figure 4 and Figure 5 show a comparison between the TCAD simulation and experimental results of the 22 nm FDSOI device. From the figures, it can be seen that when the drain voltage was 0.1 V, the simulation results of the PMOS and NMOS were larger than the experimental results. This was mainly because the surface roughness of the device affected the mobility of the device, resulting in the leakage current of the device being less than the theoretical value. When the drain voltage was 0.8 V, the simulation results of the PMOS and NMOS were basically consistent with the experimental results. This was mainly because the device operated in the saturation region, and the channel length of the device was shorter. As the gate voltage increased, the effective channel length of the device became shorter, leading to an increase in the drain current of the device and improving the impact of nonideal effects such as mobility, resulting in the simulation results of the device becoming closer to the actual results. Comparing the threshold voltage parameters with the experimental data, the threshold voltage errors of the NFET and PFET devices were 1.82% and 5.05%, respectively, which were less than 10%. In conclusion, the electrical characteristic parameters of the built simulation model device satisfied these requirements. The final determined parameter information of the device is shown in Table 1.

## 3. Influence of TID Effect on FDSOI MOS Devices

In order to clarify the physical model for simulating the TID effect, from the perspective that the radiation dose process affects the device, different radiation doses generate different amounts of positive oxide layer capture charges and interface trap charges within the oxide layer. Therefore, the impact of the total ionization dose (TID) on the device can be evaluated by using fixed charges and interface traps [9]. From the results of the impact of the irradiation dose on the device, it can be seen that the captured charges in the oxide layer and interface trap charges generated by irradiation dose directly affect the electron concentration in the gate oxide layer, thereby changing the drain current and charge collection efficiency.

Sentaurus TCAD is software specifically designed for simulating a semiconductor device, and it is used to simulate and explore the internal concentration, mobility, and electric field distribution of charge carriers in devices. The use of electrical parameters as a solution for semiconductor devices needs to be considered in Poisson’s equation and the carrier continuity equation. Because many parameters will be affected by numerous physical models during the solution’s process, such as mobility, device concentration, and work function, some physical models need to be used to ensure the accuracy of the simulation’s results of the characteristics of the FDSOI devices. The physical models for simulating the TID effect include mobility models, power generation composite models, thermodynamic models, irradiation models, and other simulation models. Among them, the mobility model is used to describe the mechanism by which factors, such as doping concentration and electric field affect particle mobility, make it’s characteristics become closer to the actual device’s. The generative recombination model was used to describe the basic recombination mechanism of the particles and the related recombination behavior of the collision ionization, mainly describing the carrier exchange process of the valence and conduction bands. The thermodynamic dynamics model extends the drift diffusion method to explain the electrothermal effect, and it solves the lattice temperature equation, except for the Poisson’s equation and the carrier continuity equation. The radiation model was used to simulate the effects of the different radiation exposure, radiation time, and irradiation of the transistors. Because the radiation changes the electrical performance of the electronic devices, it greatly alters the drain current and charge collection efficiency. From the perspective of irradiation, the impact of the total ionization dose (TID) can be evaluated by calculating the generated fixed charges and interface traps.

The Sentauraus TCAD includes several radiation models, but they are not fully applicable. However, because TCAD simulation software cannot directly solve equations for insulator materials (such as oxides), it is necessary to perform material transformation on this model, and an “OxideAsSemiconductor” module has been specially designed in TCAD, which can assume that the insulator has an electron hole transmission characteristics of a semiconductor. By modifying the material of a buried oxygen layer, silicon dioxide, in the “OxideAsSemiconductor” module, an irradiation-related model can be added to simulate the total dose effect of irradiation on the device. The added irradiation model is as follows.

(1)Radiation model

This model relates the volume charge to the absorbed dose. However, this software cannot directly solve the equation of insulator materials (such as oxides) [10]. Therefore, it is necessary to carry out material transformation for this model. The OxideAsSemiconductor material is specially designed in TCAD to provide the insulator with electron hole transmission characteristics of a semiconductor. The total dose effect of irradiation on the device can be simulated by modifying the material, silicon dioxide, of a buried oxygen layer in a OxideAsSemiconductor module, and adding a relevant model of irradiation to it.

(2)Trap model

The trap model can be used to reproduce the existence of traps on the interface. The setting of the trap concentration needs to consider the absorbed dose [10].

### 3.1. Physical Models

The radiation model used in the study is as follows.
(1)$$G_{r}=g_{0}\cdot dD/dt\cdot Y(E)$$
(2)Y(E)=(E+E0E+E1)m
where $G_{r}$ is the number of holes without initial recombination; $g_{0}$ is the number of electron–hole pairs per unit volume of material after absorbing 1 rad radiation energy; ${SiO}_{2}$ material is $7.6\times 10^{12}ehp/(rad\;{cm}^{3})$; D is the total irradiation dose; and dD/dt is dose rate (rad/s). Y(E) is the hole yield, which is related to the electric field; $E_{0}$ is the constant 0.1 V/cm, $E_{1}$ is the 0.55 MV/cm; and the constant, m, is 0.7 when the irradiation source is Co-60 [10].

Because of the high electron mobility, the carriers in the oxide will be swept out within a few picoseconds after generation and collected on the corresponding electrode. Compared with the mobility of the electrons, the mobility of the holes is much lower. Because of the slow jump propagation of the holes, the probability of the holes being captured near the interface is greatly increased. Because of the tunneling mechanism, some holes are also swept into the top layer of the silicon. The remaining holes in the oxide volume can be calculated using the fraction function of Formula (2), which depends on many processing parameters and the applied electric field. The trapped holes in the oxide layer form the fixed oxide charge component of the ionization damage. In order to simulate the total fixed charge in the irradiated oxide, if the oxide is thick enough (more than 15 nm to avoid the tunneling effect), the total oxide charge concentration caused by the radiation can be expressed as the following formula [11].
(3)$$N_{ot}=g_{0}\cdot D\cdot Y(E)$$

Sentaurus TCAD includes this model. Refer to the relevant literature and adjust the parameters, where the generation rate of the electron–hole pair is $g_{0}=7.6\times 10^{12}\;{rad}^{-1}\cdot {cm}^{-3}$, $E_{0}=0.1\;V/cm$, and m=0.9.

After the electron–hole pair is generated with ionizing radiation, the electrons are swept out of the oxide layer, and the remaining holes are transported slowly inside. The shallow level intrinsic defect oxygen vacancy ($V_{O\delta }$) in the oxide layer easily captures the hole and forms the metastable field center, $E_{\delta }\prime$. The center constantly captures and releases the hole, causing the hole to exchange among the different oxygen vacancies. The oxygen vacancy defect level near the $Si/{SiO}_{2}$ interface is deep, and the trapped hole easily generates the stable $E_{\gamma \prime }$ center. The TCAD numerical simulation method usually measures and characterizes the ability of the defects to capture electrons or holes by capturing the cross-section ($\sigma_{n}$,$\sigma_{p}$), and $\sigma_{n},\sigma_{p}$ is defined as the target region, where electrons or holes are captured by defects. The capture cross-section is related to the concentration, energy level, and spatial distribution of the defects. With reference to the relevant literature, the defect distribution type, energy level position, concentration, and size of the cross-sections of the trapped electrons or holes in ${SiO}_{2}$ are set quantitatively. In the traps model, the hole trap concentration in the layer is defined as $1\times 10^{18}{cm}^{-3}$, and the capture cross-sections of the holes and electrons are $1\times 10^{-14}{cm}^{2}$ and $1\times 10^{-16}{cm}^{2}$. The neutral hole traps do not show electricity when no holes are captured, and they are positively charged after the holes are captured [12].

In the simulation, the mobility of the electrons and holes generated by irradiation is set to a constant value of 20 cm^2^/V·s and 1 × 10^−5^ cm^2^/V·s.

The simulation model in Sentaurus TCAD is specifically expressed as follows.

Physics (Region = “R.Box”) {

Radiation(DoseRate = @doserate@ DoseTime = (0.2,@ < dosetime + 0.2 > @) doseTsigma = 0.2)

Traps(ElectricField hNeutral Conc = 1 × 10^18^ EnergyMid = −0.5 FromMidBandGap eXsection = 1.0 × 10^−14^ hXsection = 1.0 × 10^−16^ eJfactor = 0 hJFactor = 0)

}

### 3.2. Simulation Set-Up

Physical models simulating the TID effect include the mobility, generation recombination, thermodynamic, and radiation models.

In the TCAD simulation model, simulations of γ-ray irradiation with 0 k, 100 k, 300 k, 500 k, and 800 k rad (Si) were performed on the aforementioned types of devices in the TID experiments. The transfer characteristic curves of irradiation are shown in Figure 6 and Figure 7. The results show that the threshold voltage of the NMOS device drifts in the negative direction with an increase in the radiation dose, and the threshold voltage of the PMOS device also drifts in the negative direction with an increase in the radiation dose. Compared with the irradiation curve of the FDSOI devices with a buried oxygen layer thickness of 28 nm [13], the simulated transfer characteristic curve changes, and the threshold voltage drift is basically consistent with the trend and range of the changes in the reference literature.

The maximum transconductance method was used to extract the curve parameters. The variation in the threshold voltage drift with the irradiation dose under different irradiation doses was obtained, as shown in Figure 8. The results show that the threshold voltage of the NFET decreased, and the absolute value of the threshold voltage of the PFET increased with an increasing radiation dose. This is because the external diffusion of the oxygen atoms in the oxide layer did not match the surface lattice; therefore, there were a large number of oxygen vacancies near the $Si/{SiO}_{2}$ interface, which acted as trap centers. Since the transport speed of a hole is slower than that of an electron, the probability of the hole capture was greater, which forms a positively charged trapped hole [14], as shown in Figure 9 and Figure 10. With an increase in the irradiation dose, the number of trapped charges also increased in the buried oxygen layer (BOX layer), which is directly below the channel. Taking NMOS as an example, the hole trap charges captured by the BOX layer were mainly distributed below the channel, and as the irradiation dose increased, the concentration of the trap charges in the BOX layer became increasingly higher. These traps that capture holes display positive electrical properties, and the formation of positive oxide layer trap charges at the Si-SiO_2_ interface; these trapped charges change the potential of the interface between the top silicon and BOX and further affect the potential of the top silicon front gate through the coupling effect between the front and rear channels, causing a drift in the threshold voltage.

### 3.3. Experience Test

In order to verify the accuracy of the simulation model, the irradiation experiment relied on a Co-60 γ-ray source from the Xinjiang Institute of Physical Chemistry to irradiate NMOSFET devices with a gate length of 22 nm and a gate width of 160 nm, as well as PMOSFET devices with a gate length of 22 nm and a gate width of 220 nm. The sample was placed in a closed environment with an irradiation source, the irradiation dose rate was set to 150 rad (Si)/s, and the electrical characteristics of MOSFETs were measured at a total irradiation dose of 300 krad (Si), 500 krad (Si), and 800 krad (Si). This was used to compare and study the degradation degree of the different dose devices and the accuracy of the simulation models.

Because of the main impact of irradiation on the device, the electrical parameter was the threshold voltage, so it was necessary to test the transfer characteristic curve of the MOS devices in the irradiation experiment, and the transfer characteristic curve testing conditions were as follows.

(1)NMOSFET

The drain voltage was maintained at 0.1 V, the gate voltage (Vg) was scanned from 0 to 1 V, the source was grounded, and the change in drain current (Id) with the gate voltage was tested.

(2)PMOSFET

The drain voltage (Vd) was maintained at −0.1 V, the gate voltage Vg was scanned from −1 to 0 V, the source was grounded, and the variation of the drain current (Id) with the gate voltage was tested.

Because of the degradation of the effect of the irradiation dose on the device over time, the testing process of the sample MOSFET was carried out within one hour of the completion of the irradiation experiment when the irradiation dose reached the preset dose point, and the same B1500 semiconductor tester was used for the measurement. The TCAD simulation results and experimental testing results of the transfer characteristics changes of the NFET devices and PFET devices under different irradiation doses are shown in Figure 11. To analyze the accuracy of the established device model and the total dose model and to compare the changes in the electrical parameters under different irradiation doses, further extraction of the variations in the electrical parameters of the device with the irradiation dose using MATLAB software was conducted. Figure 12 shows the validation of the threshold voltage parameters of the device. The maximum error of the NFET device simulation model compared to the actual test device’s threshold voltage is 1.98%, and the PFET device simulation model compared to the actual test device’s threshold voltage error is 9.92%. In summary, the electrical characteristic parameters of the built device simulation model meet the accuracy requirements.

## 4. Influence of TID Effect on Inverter Circuits

The analysis in the third section shows that the threshold voltage of the NMOS device continuously drifted in the negative direction with the increase in the radiation dose, and its voltage clamping ability became worse, which may have caused a false touch or even loss of the switching characteristics in the circuit. The threshold voltage of the PMOS device also drifted in the negative direction with an increase in the radiation dose; that is, it required a larger reverse voltage to turn on. These changes can lead to circuit dysfunction and failure.

Inverter circuits are widely used as the most basic CMOS circuit structures. Therefore, studying the total dose effect of CMOS circuits based on inverters is important [15]. Currently, the latest modeling method for research on circuit level total dose effects is to calibrate Berkeley Short-Channel IGFET Model Level3 (BSIM3) model parameters based on Simulation Program with Integrated Circuit Emphasis (SPICE) software to replace irradiated device models and, based on this, simulate the impact of total dose effects on circuits at different irradiation doses [16]. Although the simulation speed of this method is fast, the accuracy of the BSIM3 simulation method needs to be improved, because it is only aimed at conventional silicon based MOS devices and cannot be accurately used for the simulation of nanoscale FDSOI devices. In summary, this study was based on physical device modeling and used a hybrid simulation method to simulate and analyze the TID effect of inverter circuits. It can comprehensively consider the impact of the total amount of the effect on the electrical parameters, such as the threshold voltage, mutual coupling capacitance, and parasitic resistance, of devices in the circuit, so as to conduct a more accurate results analysis of the logic threshold, noise tolerance, signal delay, and other parameters of an inverter under irradiation conditions.

The inverter structure is shown in Figure 13. In the device structure, NMOS and PMOS are calibrated devices. Based on Sentaurus software, a hybrid mode simulation method was used to add an irradiation model to the circuit to simulate the total dose effect on the inverter. Figure 14 and Figure 15 show the voltage–transfer curve (VTC) and voltage–gain curve of the inverter, respectively. The results show that the VTC shifted to the left with an increase in the irradiation dose. To quantitatively analyze the influence of the total dose on the inverter circuit, we processed the data on the curve. Figure 16 shows the logic threshold (V_M_) for the different radiation doses, where V_M_ is the voltage value corresponding to the input and output points in the transmission characteristic curve. The noise tolerance of the logic high level (N_MH_) and low level (N_ML_) refer to the tolerance range for the correct reception of the logic high- and low-level signals, respectively. Figure 17 shows the variation in the noise margins (N_MH_ and N_ML_) at different radiation doses. The results show that with an increase in the radiation dose, the logic threshold voltage of the inverter decreases, the high-level noise tolerance range increases, and the low-level noise tolerance range decreases. This is due to the traps generated by irradiation in the BOX layer, which capture holes at the Si-SiO_2_ interface and form positive trap charges. Thus, it affects the potential of the top silicon and causes the threshold voltage of the NMOS device to decrease with an increase in irradiation dose. Similarly, the absolute value of the threshold voltage of the PMOS device increases with an increase in irradiation dose. This is also the reason for the negative drift in the voltage transmission curve of the inverter and reduction in the logic threshold. The results show that the logic threshold shift of the inverter was approximately 0.052 V when the irradiation dose was 400 krad (Si).

## 5. Conclusions

Because of the pursuit of the development of Moore’s law, the traditional bulk silicon process has been found to meet the requirements of small devices with difficulty. FDSOI technology has achieved full dielectric isolation and has greater advantages in terms of speed, integration, and radiation resistance [17]. Therefore, FDSOI technology, which is the basic device process of integrated circuits, has attracted increasing attention in many fields, particularly in aviation and other fields. Its radiation resistance performance has also received increasing attention. Therefore, the study of inverter circuits based on FDSOI CMOS devices is significant for the subsequent application of FDSOI technology in integrated circuits.

The total ionizing dose effect was found to change the threshold voltage of the MOS devices. With an increase in the radiation dose, the threshold voltage of the NMOS decreased, and the absolute threshold voltage of the PMOS devices increased. In particular, when the irradiation dose was 300 krad, the threshold voltage drift rate of the NMOS device reached 11.72%, which exceeded 10%, and the device was judged to have failed. When the irradiation dose was 700 krad, the threshold voltage drift rate of the PMOS device reached 10.87%, which exceeded 10%, and the device was judged to have failed.

In addition, a simulation of the total ionizing dose effect of the inverter was performed. With an increase in the radiation dose, the voltage transmission curve of the inverter drifted negatively, the logic threshold value decreased, the high-level noise tolerance range increased, and the low-level noise tolerance range decreased. When the irradiation dose was 400 krad (Si), the logic threshold shift of the inverter was approximately 0.052 V. Compared to the inverter input voltage of 0.8 V, it generated a drift of 6.5%. The low-level range of the digital circuits was 10% of the high-level signal, so the impact of the irradiation dose on the inverter cannot be ignored.

## Figures and Tables

**Figure 1 micromachines-14-01438-f001:**
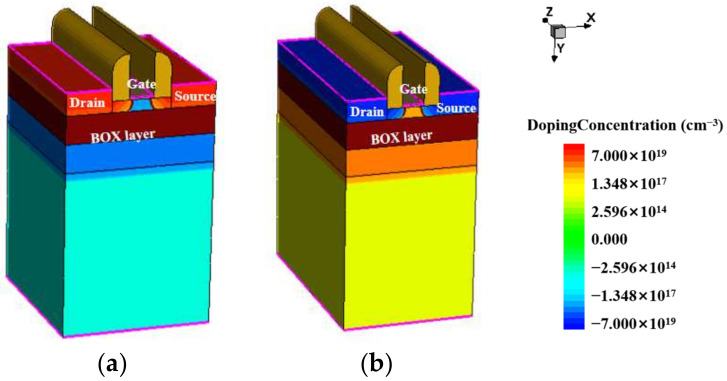
Calibration model: structure of (**a**) NMOS and (**b**) PMOS.2.2.

**Figure 2 micromachines-14-01438-f002:**
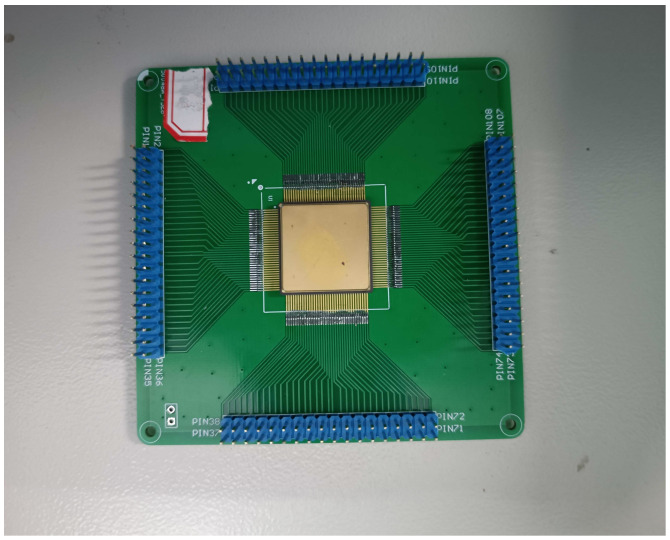
The test device diagram.

**Figure 3 micromachines-14-01438-f003:**
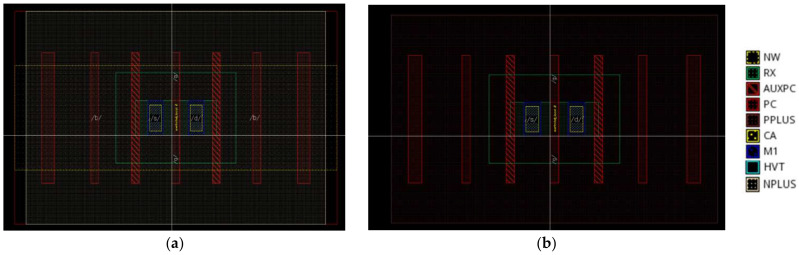
Layout file: (**a**) NMOS; (**b**) PMOS.

**Figure 4 micromachines-14-01438-f004:**
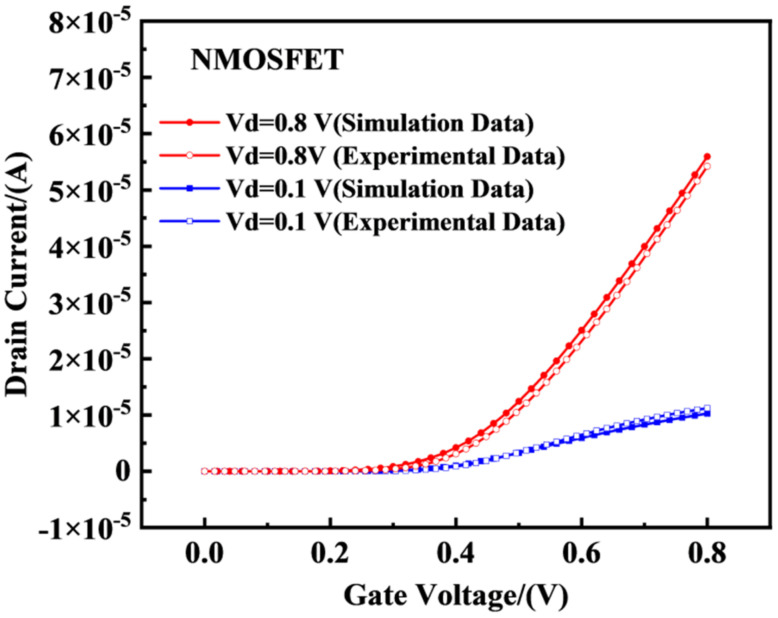
Comparison of the experimental transfer characteristic curves of the NMOS devices. When simulating irradiation, the bias was set as Vds = 0.1 V/0.8 V.

**Figure 5 micromachines-14-01438-f005:**
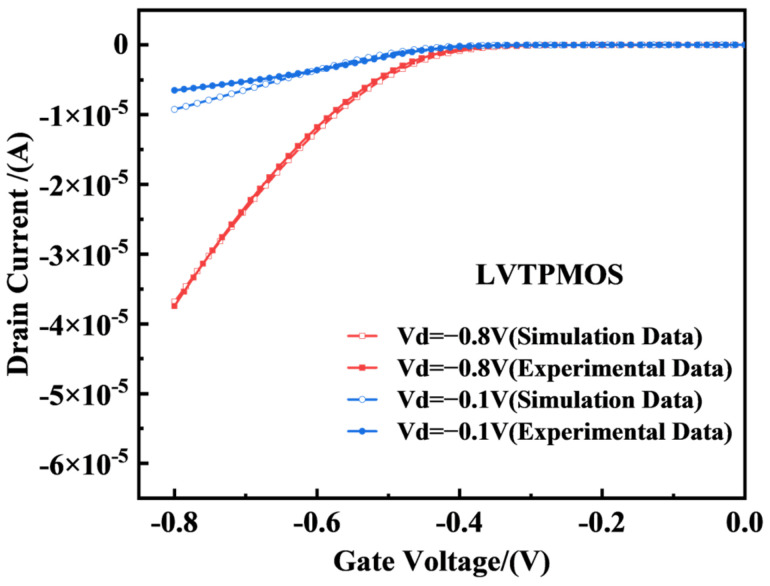
Comparison of the experimental transfer characteristic curves of the PMOS devices. When simulating irradiation, the bias was set as Vds = −0.1 V/−0.8 V.

**Figure 6 micromachines-14-01438-f006:**
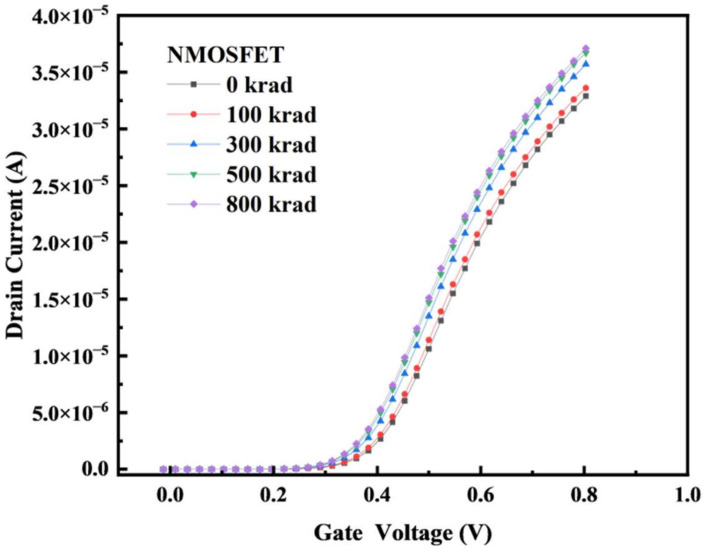
When simulating irradiation, the bias was set as Vds = 0.1 V. The I–V curves are plotted before irradiation and at several TID steps (gray curves) up to 800 k rad (Si).

**Figure 7 micromachines-14-01438-f007:**
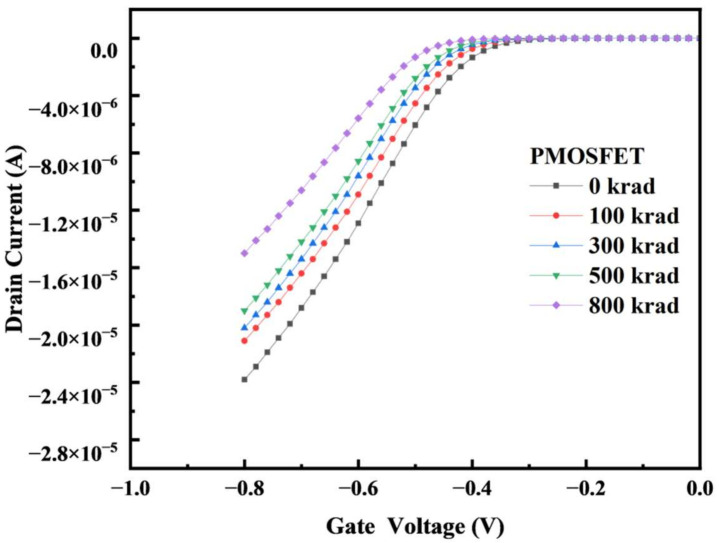
When simulating irradiation, the bias was set as Vds = −0.1 V. The I–V curves are plotted before irradiation and at several TID steps (gray curves) up to 800 k rad (Si).

**Figure 8 micromachines-14-01438-f008:**
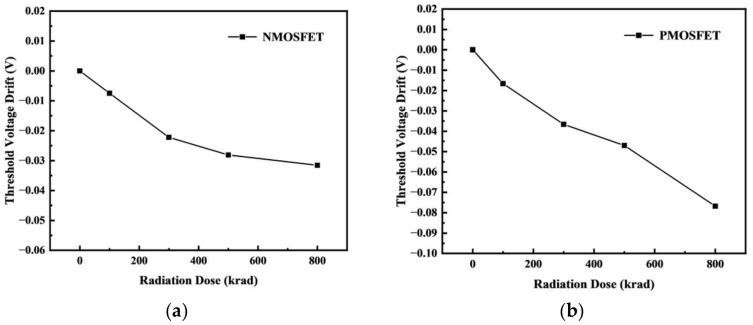
Variation in the threshold voltage drift under different irradiation doses: (**a**) NMOS; (**b**) PMOS.

**Figure 9 micromachines-14-01438-f009:**
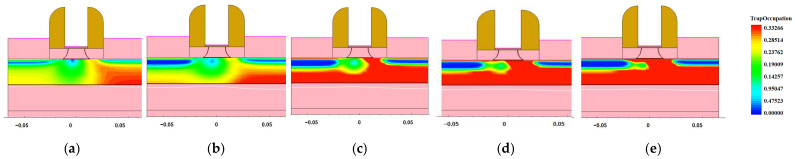
The distribution of trapped charge in the BOX layer of NMOS devices after dose irradiation: (**a**) 100 krad; (**b**) 300 krad; (**c**) 500 krad; (**d**) 700 krad; (**e**) 1 Mrad (Si).

**Figure 10 micromachines-14-01438-f010:**
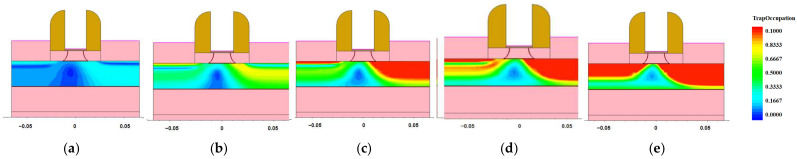
The distribution of trapped charge in a BOX layer of PMOS devices after dose irradiation: (**a**) 100 krad; (**b**) 300 krad; (**c**) 500 krad; (**d**) 700 krad; (**e**) 1 Mrad (Si).

**Figure 11 micromachines-14-01438-f011:**
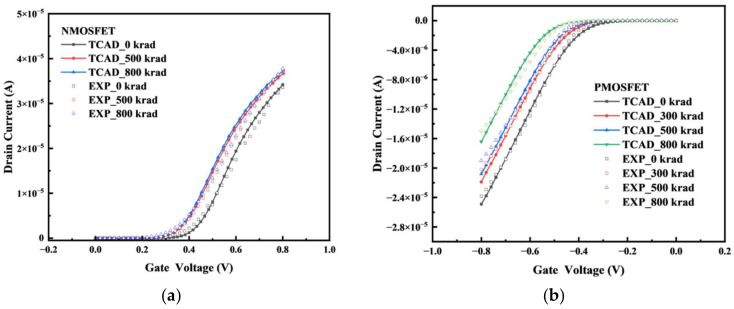
Transfer characteristics of the experiments and simulations: (**a**) NMOS; (**b**) PMOS.

**Figure 12 micromachines-14-01438-f012:**
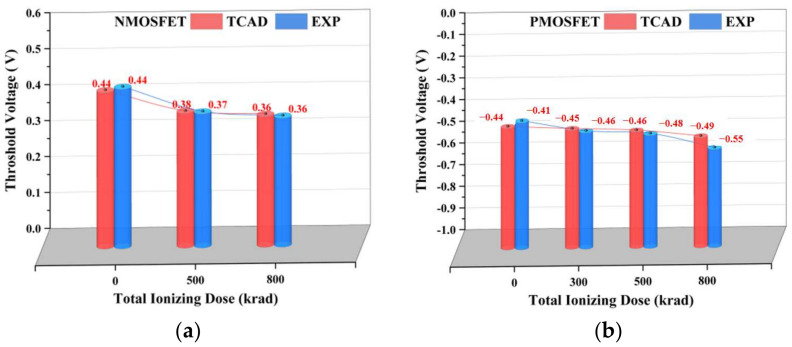
Variation in the threshold voltage of the experiments and simulations: (**a**) NMOS; (**b**) PMOS.

**Figure 13 micromachines-14-01438-f013:**
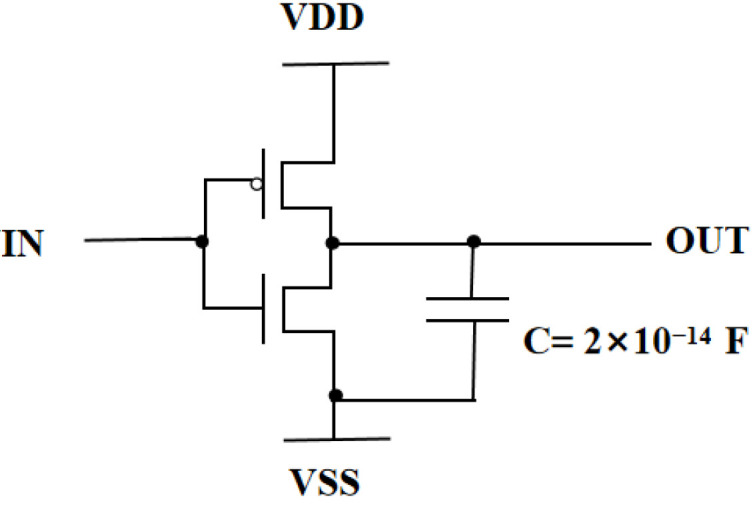
The inverter structure.

**Figure 14 micromachines-14-01438-f014:**
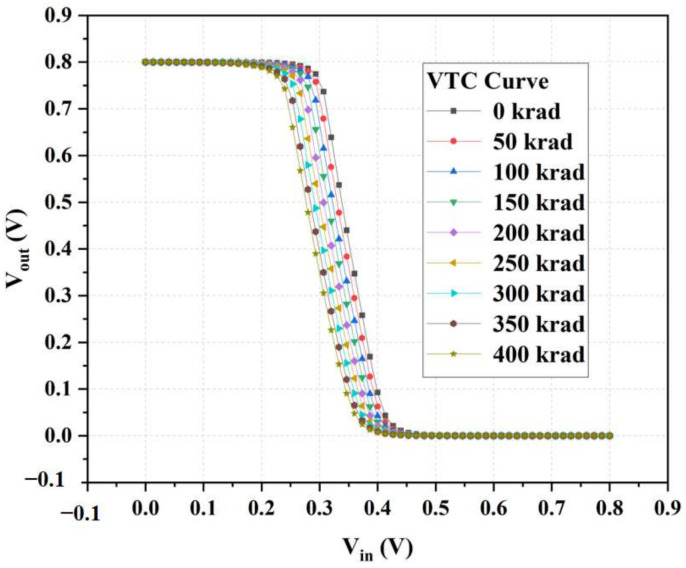
Voltage–transfer curve (VTC) of the inverter for different total ionizing doses.

**Figure 15 micromachines-14-01438-f015:**
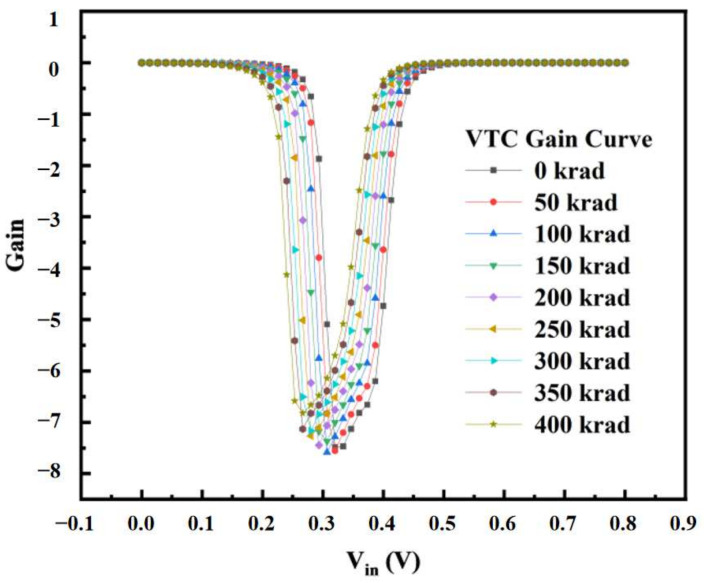
The voltage–gain curve of the inverter for different total ionizing doses.

**Figure 16 micromachines-14-01438-f016:**
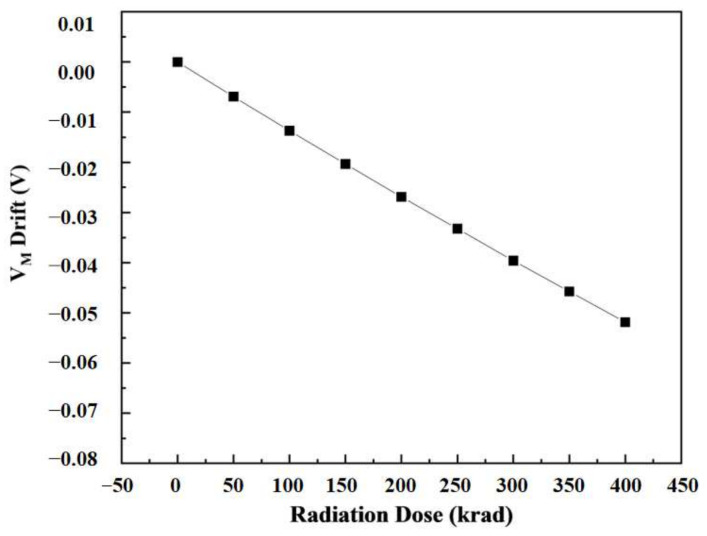
The logic threshold (V_M_) of the inverter for different total ionizing doses.

**Figure 17 micromachines-14-01438-f017:**
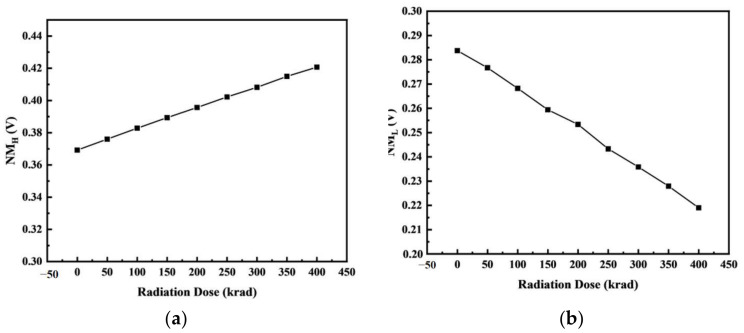
Noise tolerance of the (**a**) logic high level (N_MH_) and (**b**) low level (N_ML_) of the inverter for different total ionizing doses.

**Table 1 micromachines-14-01438-t001:** Parameters of the 22 nm FDSOI device model.

Parameter	NMOSFET	PMOSFET
Gate length (nm)	22	22
Width of device (nm)	160	220
Length of drain (nm)	40	40
Width of drain (nm)	10	10
Thickness of BOX layer (nm)	35	35
Thickness of well (nm)	25	25
Thickness of substrate layer (nm)	146	146
Thickness of gate oxide layer (nm)	2	2
Drain doping/Ldd doping(cm^−3^)	7 × 10^19^/7 × 10^18^	7 × 10^19^/7 × 10^18^
Substrate doping (cm^−3^)	1 × 10^15^	1 × 10^15^
Well doping (cm^−3^)	2 × 10^18^	2 × 10^18^
Work function (eV)	4.50	4.68

## Data Availability

Not applicable.

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
