# Peer review of "Simulation of Total Ionizing Dose Effects Technique for CMOS Inverter Circuit"

_micromachines, 2023, doi:10.3390/mi14071438_

Round 1
Reviewer 1 Report (New Reviewer)

Some fragments are disorder and have very dark meaning
Author Response
Thank you for your suggestion. We are very sorry for the improper description in the original manuscript due to our poor English writing skills. The manuscript has been revised, though there might still be some mistakes we haven’t discovered due to the time limit. Once again, we are very sorry for the inconvenience of reading due to our poor English writing skills. Please refer to the attachment for specific modifications.

Reviewer 2 Report (New Reviewer)
This study focuses on analyzing the effects of total ionizing dose (TID) on a CMOS inverter circuit based on 22 nm fully depleted silicon on insulator (FDSOI) transistors. While most previous research has focused on single event effects, the TID effect has received little attention.
The researchers first constructed and calibrated N-type FDSOI metal-oxide semiconductor (NMOS) and P-type FDSOI metal-oxide semiconductor (PMOS) structures. They studied the transfer characteristics and trapped charge distribution of these devices under different irradiation doses.
Next, the TID effect on an inverter circuit composed of these two MOS transistors was examined. The simulation results indicated that when the radiation dose reached 400 krad (Si), the logic threshold drift of the inverter was approximately 0.052 V. These findings provide insights into the impact of TID on integrated circuits in radiation environments. It recommends for publication. Comments as follows:
1. The components in Figure 3 are unclear. It is recommended to include a zoom-in view to improve clarity.
2. Could you please indicate the number of samples measured for Figures 4-8 and 12-17? If the figure includes data from multiple samples, kindly include the sample size in the figure captions.
3. Please format references.
It recommends further polish English by native speakers.
Author Response
Thank you for your suggestion. We are very sorry for the improper description in the original manuscript due to our poor English writing skills. The manuscript has been revised, though there might still be some mistakes we haven’t discovered due to the time limit. Once again, we are very sorry for the inconvenience of reading due to our poor English writing skills. Please refer to the attachment for specific modifications.

Reviewer 3 Report (New Reviewer)
This manuscript does not seem to contain serious errors, but it does not contain anything new or interesting either. The abundance of commonplaces and repetitions is the main complaint about this manuscript, which makes it uninteresting.
Comparison with experiment was carried out formally. The irradiation procedure itself is not described in any way.
There are also a lot of minor inaccuracies and typos in the manuscript, which makes it difficult to read.
E.g.
- Please correct in lin lines 134-135
[dD/dt] = rad/s
[g0]= 1/(rad cm^3)
- Oxide thickness in eq. 3 is missed.
Author Response
Thank you for your suggestion. We are very sorry for the improper description in the original manuscript due to our poor English writing skills. The manuscript has been revised, though there might still be some mistakes we haven’t discovered due to the time limit. Once again, we are very sorry for the inconvenience of reading due to our poor English writing skills. Please refer to the attachment for specific modifications.

Round 2
Reviewer 1 Report (New Reviewer)
Second review of the paper „ Simulation Of Total Ionizing Dose Effects Technique for 2 CMOS inverter circuit” submitted for the journal Micromachines MDPI.
The paper describes the behaviors of some electronic semiconductor elements such as: NMOS and PMOS under irradiation with different very high dose rates.
First of all the dose rates are described in the reviewed work may be observed in the cosmos for spacecrafts. After reading the second version the reviewer has several remarks as follow:
1. Similar to the first version the authors used “rad” as an unit of radiation dose, rad is not unit in SI and this unit is the absorption dose. In the paper the authors used the source Co-60, the isotope emits gamma and beta radiation and the dose with the source is an exposure dose or effective dose, so the dose should be expressed as Coulomb per kg or Sieverts.
2. Lines 168-170: here m=0.9 but at the formula 2 m=0.7 why? additionally the authors should more clearly explain the formula 3
3. At the Figure 13, the unit of capacity C should be added.
Author Response
Thank you for your suggestion. If there are any other issues with the article, please contact us further and we will actively correct them. Once again, we apologize for the difficulty in browsing the article caused by my English writing proficiency. Thank you for your comments.

Reviewer 3 Report (New Reviewer)
The value of the constant E0 on line 155 is absurdly small, especially compared to E1. This meaningless value should either be set to zero or corrected if it's an accidental typo.
Author Response
Thank you for your suggestion. If there are any other issues with the article, please contact us further and we will actively correct them. Once again, we apologize for the difficulty in browsing the article caused by my English writing proficiency. Thank you for your comments.

This manuscript is a resubmission of an earlier submission. The following is a list of the peer review reports and author responses from that submission.
Round 1
Reviewer 1 Report
In Section 2.2 the authors added more text about calibration of the devices simulated in the Sentaurus environment. That’s good but more information is still needed. Are their original FDSOI devices of regular well or flip-well type – it is not clear from the layouts. What were the bias voltages of the wells? The drawings (Fig. 1 with incorrect caption) still show different (n-type and p-type) substrates while all real FDSOI devices are on the same p-type substrate. The authors write that in order to calibrate the model they treated “doping concentration and mobility in the source and drain regions” as the fitting parameters. Do they really changed the concentration and mobility in the source and drain regions and not in the channel? In the existing FDSOI technologies the effective gate oxide thicknesses are different for NMOS and PMOS but both simulated devices have the same gate oxide thickness. To summarize, it still is doubtful whether the devices they simulated represent well the original FDSOI devices. In my opinion the results can only be treated as qualitative illustrations of the trends. It makes no sense to claim numerical results (such as threshold voltage drift rate) with four significant digits.
Reviewer 2 Report
Not at the state of the art. All the physical processes described are well known from several years.
Several other publications already are dealing on this topic with more in-depths results.
Comparison of TCAD simulations with experimental results of irradiated devices could be of interest according to what has already been published.